# Thermooptical PDMS-Single-Layer Graphene Axicon-like Device for Tunable Submicron Long Focus Beams

**DOI:** 10.3390/mi13122083

**Published:** 2022-11-26

**Authors:** Giancarlo Margheri, André Nascimento Barbosa, Fernando Lazaro Freire, Tommaso Del Rosso

**Affiliations:** 1Institute for Complex Systems of National Council of Researches of Italy, Separate Location of Sesto Fiorentino, Via Madonna del Piano, Sesto, 50019 Florence, Italy; 2Department of Physics, Pontifícia Universidade Católica do Rio de Janeiro, Rua Marques de São Vicente, Rio de Janeiro 22451-900, Brazil

**Keywords:** axicons, thermo-optical devices, beam shaping, long focusing devices, graphene–PDMS hetero interfaces

## Abstract

Submicron long focusing range beams are gaining attention due to their potential applications, such as in optical manipulation, high-resolution lithography and microscopy. Here, we report on the theoretical and experimental characterization of an elastomeric polydimethylsiloxane/single layer graphene (PDMS/SLG) axicon-like tunable device, able to generate diffraction-resistant submicrometric spots in a pump and probe configuration. The working principle is based on the phase change of an input Gaussian beam induced in the elastomer via the thermo-optical effect, while the heating power is produced by the optical absorption of the SLG. The phase-modified beam is transformed by an objective into a long focus with submicron diameter. Our foci reach an experimental full width at half maximum (FWHM) spot diameter of 0.59 μm at the wavelength of 405 nm, with the FWHM length of the focal line greater than 90 μm. Moreover, the length of the focal line and the diameter of the focus can be easily tuned by varying the pump power. The proposed thermo-optical device can thus be useful for the simple and cheap improvement of the spatial resolution on long focus lines.

## 1. Introduction

Long focusing range beams (LFRB), also known as quasi-Bessel beams (QBBs), are appealing due to their non-diffractive properties. Their use in various important applications, including scanning microscopy [1,2,3], optical manipulation [4,5] and lithography [6,7,8], would be enhanced if the focal spot had a submicron full width at half maximum (FWHM) diameter. As is well known, the submicron requirement can be fulfilled only if the device producing LFRBs has a high numerical aperture (NA). This is not an easy task to achieve with refractive axicons (refraxicons) whose NA cannot exceed 0.75 [9]; the other widespread axicon devices use a high NA microscope objective with a central obscuration that strongly lowers the efficiency.

Moreover, refraxicons have two typical fabrication errors: (i) an elliptical cross-section resulting in a nonrotational symmetric beam, and (ii) imperfections of the tip, which is typically shaped as an aspherical micro lens [10], introducing strong fluctuations in the focus line. Several techniques have been developed to bypass the effects of imperfect tips, such as beam expansion, Fourier filtering [9], the use of liquid immersion [11], laser wet etching [12] or the use of photonic crystals and metasurfaces [9,13,14]. However, these solutions are often not efficient or are technologically difficult to implement.

An alternative is the use of positive reflective axicons, with benefits including the absence of chromatism, tolerance to high power densities and a very small cuspidal region [15]. These characteristics allow working with small laser beams and minimize the intensity fluctuations in the Bessel focus [10]. However, in a reflective geometry, the focus is close to the sample surface and on the same side of the input beam, with consequent obscuration effects. In order to avoid these drawbacks, the refraxicon must work in a tilted geometry, which is quite prone to aberrations, particularly for foci with submicrometric spot diameters. Thus, when a system based on a reflecting axicon is dedicated to submicrometric focusing, it usually presents a relevant opto-mechanical complexity [16], low stability and high cost. 

Another desirable feature when working with LFRBs is the possibility to tune the light distribution without moving parts in the experimental set-up. In this case, the tuning function is generally obtained with complicated and high-cost experimental benches, or by the use of specialized components based on metasurfaces [17,18,19]. 

In this scenario, the formation of LFRBs with submicrometric cross-section is a field in which improvements are still demanded, especially in terms of optical efficiency, simplicity in the fabrication of the device and the tunability of both the focus diameter and extension. 

Here, we propose an alternative solution in which LFRBs are formed by exploiting bilayers composed of a weakly absorbing film and a thicker layer possessing thermo-optical properties. Under proper illumination, the latter film develops a spatial variation of the refractive index, which modifies in an axicon-like way the phase of a probe beam. Many organic polymers exhibit negative thermo-optical coefficients (TOC) with a high absolute value (i.e., polymetylmetacrilate (PMMA) or polystyrene (PS)). Among these materials, we found that the most favorable performances concerning TOC, transparency, durability, stability, processability, cost and ease of supply are exhibited by the elastomer polydimethylsiloxane (PDMS). For PDMS, which is stable up to 200 °C, the absolute value of the TOC is around 4.5 × 10^−4^ °C^−1^, approximately 4-fold higher than that of PMMA or PS.

The heating function is performed by the weakly absorbing film, which must also have high thermal conductivity. The choice of the proper optical absorbing layer is not a simple task. In fact, the ideal absorber should present the maximum transparency together with suitable heating behavior in the whole Vis–NIR region of the electromagnetic spectrum. A possible approach would be to consider absorbers based on metallic or semiconductors thin films, whose fabrication, however, is not an easy task using conventional techniques [20,21,22,23]. Indeed, the vapor condensation on the dielectric support may lead to a randomly nanostructured surface [21,22,23]. This is particularly problematic for noble metals thin films, where the nanostructured surface is characterized by a wavelength-dependent localized surface plasmon resonance (LSPR). While the metal nanostructures can be fruitfully used to perform numerous surface-enhanced spectroscopies [24,25], they represent a serious drawback for the performances of a wideband thermo-optical axicon. Metallic alloys based on nickel are used for the fabrication of commercial neutral optical density filters (see, for instance, Thorlabs neutral density reflective filters) up to a transmittance of 80% and an absorption of 10%, but they suffer from oxidation instability when the temperature rises up to 100 °C. Other nonmetallic absorbing substances, such as carbon black or graphite, could be used to heat the polymer, but they cannot be deposited in films thin enough to guarantee a transparency higher than 70% [26,27].

To summarize, the fabrication of ultrathin thin absorbers with (1) high flat spectral transmission, (2) sufficient absorption in the Vis–NIR spectral region, (3) high and uniform heat conduction and (4) high chemical and structural stability is currently a difficult challenge.

In this context, particular attention is focused on the emerging 2D materials, thanks to their intrinsic characteristic of small effective thickness [28] and interesting optical properties for photonics applications [29,30,31,32]. 

Among the low-dimensional materials, graphene was the first to be obtained experimentally [33,34], and now it can be synthesized by chemical vapor deposition (CVD) in a controlled number of layers and in foils having linear dimensions in the cm range [35]. These features greatly simplify the optical excitation of the 2D material, together with the precise control of the optical properties, which strictly depend on the number of single-layer graphene (SLG) foils. Indeed, in spite of its effective thickness of 0.34 nm, a typical SLG has an unusual optical absorption of 2.3%, almost flat in the Vis–NIR spectrum [36,37], which represents an unsurpassed advantage with respect to other light absorbers. Moreover, its thermal conduction is one of the highest known (4000 Wm^−1^ K^−1^), and it is expected to efficiently drain the heating power at the graphene–PDMS interface. 

In this work, we demonstrate that in spite of its low absorption, the SLG can significantly raise its temperature, even delivering moderate optical pump powers, heating the PDMS sufficiently to push its temperature close to the tolerance limit (~470 K) and generating a remarkable refractive index gradient. Moreover, the control of the transverse and longitudinal dimensions of the spot is performed optically by varying the pump power, allowing the fine-tuning of the focus diameter and its extension. 

## 2. Thermooptical and Electromagnetic Modeling 

### 2.1. Model of the PDMS/SLG Thermooptical Device

The working principle of the proposed thermooptical axicon-like device is reported in Figure 1. 

The device is based on a PDMS/SLG bilayer used in a pump and probe configuration. The pump beam (wavelength: 405 nm) is tightly focused onto the SLG, which absorbs part of the light and heats the elastomer, generating a negative gradient index (GRIN) distribution Δn. A Gaussian probe beam is launched to the device from the back side and its phase is transformed by the GRIN distribution. As the heating of PDMS is pointy, the thermo-optical phase issued to the input beam resembles that produced by a negative axicon. An objective finally focuses the modified probe beam in a tight ring, with a long focusing range and a submicrometric cross-section. 

The geometrical details of the graphene–PDMS thermo-optical device considered in the simulations are illustrated in Figure 2. 

The diameter of the PDMS disk is 5 mm, and its thickness is 2 mm. The elastomer is in contact with an SLG, deposited on a BK7 glass disk with 1 mm thickness. The whole system is surrounded by air. 

The heat source, located in the SLG, is simulated with a Gaussian heat surface power distribution (W/m^2^), with a waist diameter 2*w*_0_ = 70 μm. The heating power density is Q(r) = P_h_/(*πw*_0_^2^)exp(-(*r/w*_0_)^2^), where P_h_ is the heating power and *r* is the radial coordinate on the graphene surface. The maximum P_h_ is 20 mW, and is varied in steps of 5 mW. The calculations were performed with the COMSOL MULTIPHYSICS software 5.3a, whose database of materials provides the physical constants of PDMS and BK7, while those of graphene are taken from the available literature. As visible in Figure 2, the resulting temperature distributions T(*r,z*) in PDMS exhibit a central absolute maximum T(0,0) of 470 K corresponding to P_h_ = 20 mW, decreasing on the axis with a rate of 10 K/mW. 

Even if the data sheets of PDMS (specifically Sylgard 187) claim that the elastomer is stable until ~470 K, in order to guarantee a margin of 20 K for a safe operation of the device, we will consider as the maximum heating power the value P_h_ = 16 mW, corresponding to a temperature of 450 K (~150 Celsius). The FWHM of the radial temperature distribution is constant with P_h_ and equal to 0.086 mm, indicating a smearing with respect to the width of the heating source (0.07 mm), likely due to the high thermal conduction of graphene [37]. The temperature distribution T(*r,z*) gives rise to the variation of the refractive index of Figure 3 that follows the relationship Δn(*r,z*) = −4.5 × 10^−4^(T(*r,z*)-T_0_), where T_0_ = 293.15 K is the ambient temperature. For convenience reasons, in the following, we will consider the absolute value of Δn(*r,z*) instead of the actual negative value. As anticipated, the distribution |Δn(*r,z*)| resembles a tipped structure that, in analogy with classical axicons, is responsible for the generation of annular beams (see next section). The apex region, reported in the inset, has a smoothed apex with a diameter of about 106 μm, higher than the beam waist of the focused pump beam. 

The variation of |Δn(0,0), that is, the absolute maximum of |Δn(*r,z*)|, is linear with P_h_, as illustrated in Figure 3b, with a growth rate of 0.004 refractive index units (RIU)/mW, and reaches its maximum of 0.065 at P_h_ = 16 mW. 

The radial behavior of |Δn| at *z* = 0 mm, namely |Δn(*r*,0)|, is shown in Figure 3c. Following the trend of the central maximum, at a given radius r, it decreases proportionally, with P_h_ halving at 0.067 mm from the center. 

The axial behavior of |Δn| with *z* is represented in the plot of |Δn(0*,z*)| reported in Figure 3d. Its maximum |Δn (0,0)| halves at a depth *z* = 0.062 mm from the graphene layer. The behavior of |Δn(0,*z*)| is linear with P_h_ at a given *z*-coordinate, as it happens for the r-coordinate.

Given the linear relationship between |Δn(*r,z*)| and T(*r,z*), the temperature distribution resembles the distribution of |Δn(*r,z*)|, and it has not been reported. 

### 2.2. Electromagnetic Modeling 

As anticipated, the device exploits the phase variation of a probe beam that travels through the heated PDMS and experiences a thermo-optical phase variation. A further phase change is added by a focusing objective to generate the real LFRB. These variations of the phase of the probe electric field along its propagation are schematically reported in Figure 4.

We assume that the input electric field has a Gaussian distribution E_0_(*r*_0_) = E_amp_ exp(-(*r*_0_*/w*_0_)^2^) where *r*_0_ is the radial coordinate at the input plane and *w*_0_ is the beam waist (*w*_0_ = 0.5 mm, E_amp_ = 1 V/m). Concerning the probe wavelengths, we chose two values representing approximately the range of the visible spectrum, namely *λ*_1_ = 633 nm and *λ*_2_ = 405 nm, for red and violet radiation, respectively. Considering the propagation through the heated PDMS, the electric field experiences an overall phase change ΔΦ (*r_out_*) given by:
(1)$$\Delta \Phi (r_{out})=\frac{2\pi }{\lambda }\int_{0}^{L}n(r_{1},z_{1})dl$$
where *dl* = (*dr*_1_^2^ + *dz*_1_^2^)^1/2^ is the infinitesimal path on the ray trajectory, *λ* is the wavelength of the probe beam in vacuum, *L* is the overall length of the ray trajectory and *r_out_* is the output radial coordinate. *r*_1_ and *z*_1_ are the radial and longitudinal coordinates inside the PDMS, and the refractive index of the heated elastomer is expressed as *n*(*r*_1_, *z*_1_)~*n*_0_ + Δn(*r*_1_,*z*_1_*)*, where *n*_0_ is the refractive index of PDMS at ambient temperature, and Δ*n*(*r*_1_, *z*_1_) is the refractive index change due to the thermo-optical effect. In the usual paraxial approximation, *dr/dz*_1_ < < 1, so that *dl~dz*_1_, and the integral simplifies in:
(2)$$\Delta \Phi (r_{0})=\frac{2\pi }{\lambda }\int_{0}^{s}n(r_{0},z)dz$$
where *s* is the sample thickness, *r*_0_ is the input radial coordinate and *r_out_* has been substituted by *r*_0_. 

The field at the PDMS output plane is thus given by E_0_(*r*_0_)exp(*i*ΔΦ(*r*_0_)). Then, the beam propagates towards a focusing objective whose focal length is *f*. As the objective is placed after the PDMS surface at a distance *d*~15 mm, it is located in the Fresnel zone of the output pupil (approximately represented by the field beam waist), as the Fresnel number at wavelength *λ*_1_ is *w*_0_^2^*/ld* = 26.3 >> 1 [38]. By exploiting the discussion on the Fresnel approximation, the field at the lens input plane is in good approximation of the geometrical projection of the field present at the PDMS exit face. The focusing objective issues a phase shift to the incoming field that, in the case of diffraction limited behavior, is quadratic with *r*_0_ and expressed by the well-known relationship ΔΦ_lens_(*r*_0_) = –*πr*_0_^2^*/λf*. After the lens, the field propagates towards the observation point P(*r,z*), where *r* is the distance from the optical axis and *z* is the distance from the lens. The resulting field at P is found with the application of the Fresnel integral and is calculated with the equation:
(3)$$E(r,z)=A\int_{0}^{\infty }E_{0}(r_{0})\cdot J_{0}(\frac{2\pi \cdot r\cdot r_{0}}{\lambda z})\exp(i\cdot \Delta \Phi_{lens}(r_{0}))\exp(i\Delta \Phi (r_{0}))\exp(-i\cdot \pi \frac{r_{0}{}^{2}}{\lambda z})\cdot dr_{0}$$
where the term exp(-*πr*_0_*λz*) represents the phase change due to the propagation towards the observation point, *A* = *A*_0_·exp(*ikz*)/*λz* and *A*_0_ is the amplitude of the impinging electric field. The intensity is ~|E*(r,z*)|^2^ and is expressed in normalized or arbitrary units. 

The electromagnetic modeling is used at first to calculate the effect due exclusively to the thermally induced refractive index distribution, obtained by dropping the phase contribution ΔΦ_lens_(*r*_0_) in the integral (3). The phase term ΔΦ(*r*_0_) is evaluated by considering the refractive index distribution calculated in the thermo-optical model without approximations. 

The theoretical far field intensities for the wavelength λ_1_ are shown in Figure 5. The central region of the intensity pattern gradually depletes in response to the increase in P_h_, and the peak to peak diameter increases as well. The axial intensity halves at P_h_ = 2.4 mW (Figure 3a) and vanishes at P_h_ = 4.2 mW (Figure 3b). For higher powers, the probe beam propagates into the far field as an empty annular beam (Figure 3c), reaching an angular semi-aperture of 0.0037 rads at P_h_ = 16 mW. 

The results obtained for the probe wavelength *λ*_2_ are similar, because the induced refractive index gradient does not have significant dispersion in the visible range, because the dominant effect is dependent on the temperature T [39]. 

The phase-modulated beam is in turn transformed by the focusing lens (objective), and the corresponding normalized axial and transversal intensity distributions are calculated by solving the integral of Equation (3) for the three pump powers P_h_ = 4 mW, 8 mW and 16 mW and considering the focal length *f* = 2.6 mm of a commercial objective (see Section 3). 

The results obtained for the probe wavelength *λ*_1_ are reported in Figure 6a,b, while Figure 6c,d reports the normalized intensities for the wavelength *λ*_2_.

Starting from the probe at *λ*_1_, the axial distributions show fast rises followed by slower declines, reaching absolute intensity maxima at coordinates z_max_. A weak low spatial frequency modulation is present, while the high-frequency oscillations near z_max_ are absent, demonstrating that the effect of the tip curvature (see Figure 3) is negligible [10]. The axial coordinate z_max_ has a right shift of about +20 μm passing from P_h_ = 4 mW to P_h_ = 16 mW. At P_h_ = 16 mW, the axial light distribution has a FWHM value (hereafter FWHM_line_) of 88.2 μm, and lowers significantly at lower P_h_, down to 11.5 μm (Table 1). The radial intensity distributions are calculated at the coordinate z_max_ corresponding to each heating power, and are plotted in Figure 6b. The curves show that their FWHMs (hereafter D_F_) decrease at increasing P_h_, and reach the minimum value D_F_ = 0.96 μm at P_h_ = 16 mW. Thus, the theoretical model predicts that for the red probe radiation at *λ*_1_, D_F_ is submicrometric at the maximum P_h_, while at lower heating powers, it enlarges by approximately 25% (see Table 1), reaching a value of 1.32 μm at the minimum P_h_ = 4 mW. 

The long focusing feature is evident by comparing the FWHM_line_ of LFRB with the depth of focus obtained with conventional Gaussian beams, which is routinely considered as the double of the Rayleigh range, 2*πw*_0_^2^*/λ* (hereafter R_G_). For instance, considering a Gaussian beam with a waist equal to the half of the D_F_ calculated for P_h_ = 16 mW, 0.48 μm, it results in R_G_ = 2.3 μm, which is 24 times lower than FWHM_line_ (see Table 1). Passing to the probe at wavelength λ_2_, the trend of the axial light distributions (Figure 6c) follows approximately the same behavior of those calculated for *λ*_1_, showing that the FWHM_line_ is relatively wavelength-independent (Table 1), while D_F_ decreases with it. Considering, for instance, the maximum P_h_ = 16 mW, we have D_F_ = 0.56 μm, against D_F_ = 0.96 μm at *λ*_1_. The Gaussian depth of focus corresponding to D_F_/2 = 0.28 μm results in R_G_ = 1.23 μm, 73.4 times lower than FWHM_line_. 

For the sake of clarity, the behavior of FWHM_line_ vs. P_h_ for *λ*_2_, the more favorable condition, is resumed in the 2-Y plot of Figure 7, showing an almost linear trend with a rate of growth of 7.5 μm/mW. 

The 2D intensity plots of the focal regions are shown in the false color plots of Figure 6e. The focusing of the incoming light crowns into annular light distributions appears evident (see Figure 5), in agreement with the results of the ray tracing simulation presented in Figure 1.

It is worth noting that as the focus line lengthens at increasing P_h_, the side lobes of the radial intensity shrink towards the center, subtracting power to the central lobe, whose D_F_ decreases at the same time. This last effect, however, is not sufficient to compensate for the loss of intensity due to the spread of the optical power in the side lobes, and as a consequence, the maximum intensity of the central spot decreases. This consideration is clearly evidenced in the plot of Figure 7, calculated for *λ*_2_ = 405 nm. Here, the central intensity shows a decrease of approximately a decade passing from P_h_ = 1 mW to P_h_ = 16 mW, while the FWHM_line_ expands approximately with the same ratio. As the PDMS/SLG behaves in a negative axicon-like way, similar trends are expected for a real axicon coupled to a positive lens [40]. The above result suggests that the operation with this device requires a preliminary tradeoff between the requirement of a large FWHM_line_ and a proper power concentration in the focus. In this respect, the tunability of the presented device can be useful to finely adjust the proper operational conditions.

## 3. Experimental Section

### 3.1. Fabrication of the Device

The predicted behavior was experimentally verified using a sample of SLG deposited on glass and covered with PDMS. The SLG was synthesized by chemical vapor deposition (CVD) and deposited onto BK7 glass slides of 1 mm thickness by using the experimental procedures outlined in [41,42]. The details of the methods are reported briefly in the Appendix A, where we show in Appendix A the Raman spectra of the graphene deposited on the glass, confirming the presence of a high-quality SLG. A small portion of the glass surface is not covered by graphene to serve as a reference for the absorption measurements.

The graphene monolayers were covered by a 2 mm thick PDMS layer prepared using a 10:1 mixture of monomer and curing liquid. After the mixture degassing, 1 mL of the liquid PDMS was poured onto the graphene layer (the viscosity of the liquid mixture was sufficient to avoid the overflow of the liquid out of the borders of the support) and heated at 50 °C for 20 min to efficiently activate the polymerization, completed overnight in air at ambient temperature. 

### 3.2. Experimental Set-Up

We performed two different sets of measurements, the first aimed at demonstrating the formation of the predicted far field annular beams, prerequisite for the formation of LFRBs, and the second at measuring the diameter and length of the long focus line.

The scheme of the experimental apparatus is shown in Figure 8. 

In the former case, the PDMS/SLG was illuminated from the PDMS side with a focused pump beam generated by a diode pumped solid state laser (*λ*_2_ = 405 nm), whose power can be adjusted by using an optical variable attenuator (Figure 8). At the same time, the sample was illuminated from the other side with a collimated probe beam coming from either a He-Ne laser (probe at *λ*_1_) or a diode pumped solid-state laser (probe at *λ*_2_). The diameter of the probe beam was adjusted to obtain an almost input Gaussian waist *w*_0_ = 0.5 mm, corresponding to an intensity waist of 0.36 mm, while the focus of the pump beam was adjusted in a spot with an average diameter of 70 μm. The probe beam propagates through the PDMS and is transformed into an annular ring that is projected onto a screen placed at 350 mm distance, as illustrated in the inset of Figure 8. 

The images were acquired with a CCD camera (Watec 902H) and elaborated by using the software IMAGE PRO Plus, providing in real time the light distribution on a chosen scan line during the pump power changes. After a calibration step, the measurement of the beam D_F_ can be easily performed. 

In the latter case, the measurements on the submicron LFRB were performed using the complete optical system shown in Figure 8. While the generation of a phase-modulated beam at the PDMS output remains the same as in the previous paragraph, a high NA objective Obj_1_ (Newport MVC-60x, 0.85 NA, 60x, focal length 2.6 mm), whose first interface is placed at a ~15 mm distance from the sample, focuses the light coming from the output surface of PDMS. The objective Obj_1_ is mounted on a three-axis piezo translator (Piezojena Tritor 100 piezo stage working in closed loop operation, with 80 μm translation range and 0.53 μm/V resolution), and moved in steps of 5.8 μm.

The focal spot produced by Obj_1_ is magnified by a two-stage optical system, formed by a second Newport MVC-60x objective Obj_2_ (0.85 NA, 60x, focal length: 2.6 mm) with 110x magnification, and a further 20x Galilean telescope. Thus, the transverse light distribution is magnified 2200 times, and then it is projected onto a screen placed at a fixed position, where it can be reliably observed by a CCD camera. The image was then acquired and elaborated with the same procedure used for the analysis of the annular patterns. 

The position of best focus is obtained by moving Obj_1_ axially with the piezo translator until the intensity pattern on a chosen diametric line reaches the minimum radius, and the contrast with the side lobes is maximum. A similar measurement was performed at P_h_ = 16 mW with the other probe beam at wavelength *λ*_2_. 

In order to test the long focusing performance of the device, we measured the maximum light intensity on the focal line at P_h_ = 16 mW for both radiations at *λ*_1_ and *λ*_2_. This operation was performed by moving horizontally Obj_1_, causing the axial shift of the focus line in front of the magnifying system. The screen collected the magnified image of the field intensity present on the conjugated plane π_im_ (see Figure 6). We acquired the image of the spot focus for different displacements of Obj_1_, and recorded the central maximum intensity for each displacement.

## 4. Results and Discussion 

The measured absorption of the PDMS/SLG bilayer was 3.4%, higher than that reported for a free-standing SLG (2.3%) at the wavelength *λ*_1_. As this film has sufficient quality (see Section 3.1), this effect is mainly attributable to the multireflections in the bilayer, even if a minor contribution can come from some polymer residuals. To demonstrate this, we performed a simulation using a graphene refractive index of 2.7 + 1.33i [42], a monolayer thickness of 0.34 nm and a PDMS refractive index of 1.43. The calculations, carried out with the multilayer evaluation software WINSPALL, give at 45° incidence angle a reflection R = 2.3% and a transmission T = 0.94%, with a resulting absorption *A = 1 − R − T* = 3.6%, close to the measured value. 

This weak absorption of SLG is nevertheless high enough to generate the predicted transformation of the input beam into an annular beam in the far field, as shown in Figure 9. The images of the rings recorded at a distance of 350 mm from the sample (inset of Figure 8) show the gradual depletion of the central region with the increase in P_h_. At P_h_ = 5 mW, the center appears dark, in agreement with the expected theoretical value of 4.2 mW. Figure 9b reports the linear increase in the circles’ diameter (distance between the maxima) with P_h_. The diameter subtends a full angle whose rate of growth is approximately 5.3 × 10^−4^ rads/mW. This value is close for the two wavelengths, confirming the weak dispersion of the thermo-optical coefficient of PDMS in the visible range. It is worth noting that the capability to shape annular rings could be exploited to perform important functions [43], but the optimization of the working conditions would require a more detailed investigation that is beyond the scope of the present work.

The magnified (2200x) radial light distributions of the focused probe beam at *λ*_1_ are illustrated in Figure 10. The resulting spot diameters are D_F_ = 0.97 μm, 1.25 μm and 1.4 μm for the three heating powers of 4 mW, 8 mW and 16 mW, respectively. In the case of the probe beam at *λ*_2_ (Figure 11), the measured spot diameters at the same P_h_ decrease as theoretically predicted, and their values are D_F_ = 0.59 μm, 0.63 μm and 0.73 μm for the three powers, respectively, in agreement with the theoretical findings reported in Table 1.

The experimental axial intensity distributions measured at P_h_ = 16 mW for the two probes at wavelengths λ_1_ and λ_2_ are shown in Figure 12a,b, respectively. As shown, the theoretical predictions and the experimental findings agree well. In order to check the foci robustness to diffraction, we further measured the spot diameters at the extremes of the FWHM_line_, axially displacing Obj_1_ until the two positions P_1_ and P_2_, where D_F_ is ~1 μm, are locked. At the probe wavelength *λ*_1_, this happens when the distance P_1_P_2_ is equal to 7.8 μm, while at *λ*_2_ nm, this distance is equal to 68 μm, not far from the theoretical values (8.6 μm and 75 μm, respectively).

The literature on the topic of long focusing with submicron focal spots is surprisingly not hefty, and our results can be favorably compared with other ones obtained with more sophisticated tools. For instance, in [15], the use of a refraxicon fabricated with an ablation femtosecond laser allowed the theoretical formation of a long focus with a FWHM diameter down to 0.51 μm and a FWHM depth of focus of 95 μm, at a wavelength of 780 nm. The research shown in [9] reports on the performance of a metasurface-based axicon, with a FWHM focus diameter of 0.163 μm at the wavelength of 405 nm, maintained up to a measurable distance of 160 μm, while from the data reported herein, a FWHM of the focal length of ~40 μm can be evinced. Moreover, in this latter case, the transmission efficiency is strongly wavelength-dependent. Indeed, while at the design wavelength of 532 nm, the maximum experimental transmission of the device is 50%, the efficiency can become as low as 5% at red wavelengths. A similar drawback is reported in the metasurface-based axicon reported in [14], where it is demonstrated that the efficiency loss is mainly due to the geometrical structuration of the single unit cell of the metasurface, and thus inherently affects all the devices based on this approach. In that paper, a wavelength-independent spot size of D_F_ = 0.25 μm is reported, but the propagation length within which the spot diameter is unchanged is limited to 7 μm, while a FWHM axial line focus length of ~11 μm can be inferred from the reported data. 

In our case, the transmission efficiency is ~85% in the whole visible spectrum, being limited only by the transmission of the objective. Our best focusing performances, experimentally verified at a heating power of 16 mW, are a FWHM spot and depth of focus of 0.56 μm and 90.3 μm, respectively. Noticeably, as shown in Appendix A, performances similar to those reported in [13] can be reached using the focal length of 0.9 mm of a commercial GRIN rod lens. 

Moreover, it is worth noting that the value of D_F_ may not be so decisive, depending on the application. For instance, in [16], it was experimentally demonstrated that submicron features about 5-fold smaller than the FWHM spot diameter can be obtained by pulsed laser ablation, provided that the on-axis laser fluence exceeds a certain threshold. As a consequence, submicrometric features can be fabricated using spot sizes even higher than 1 μm. In this perspective, the depth of focus can play a more significant role, and our results are aligned with other more cutting-edge long focus forming systems. Moreover, when the interplay between the focus dimension and the laser fluence can make it difficult to precisely define the parameters of the fabrication process, the availability of a tunable focus may be an important tool to refine the performances of a working facility. 

The most important drawback of the presented device is the high loss of pump power when the transmitted part is not used as a probe. However, this problem can be reduced by using multilayer graphene. Indeed, thanks to the progress of the fabrication procedures, a deposition of up to eight layers has been reported [34]. The amount of absorbed optical power could be multiplied, maintaining a high transmissivity at the same time, strongly relaxing the concern of optical pump loss. 

## 5. Conclusions

In this paper, we have reported on a proof-of-concept of a single-layer graphene–PDMS long focusing thermo-optical device producing foci with submicrometric diameters. Thanks to the high concentration of the pump power in the subnanometric thickness of graphene, significant heating can be produced even with the small amount of absorbed optical pump power, causing temperature variations that give rise to a huge thermo-optical effect in PDMS, while maintaining a high optical transmission. The generated negative gradient index distribution has a spatial cuspidal-like shape, as the apex can be tightly reduced by properly focusing the heating optical pump. An input probe beam is phase-modulated in the heated PDMS and focused by an objective with short focal length, obtaining a long focus line where the spot size has submicrometric dimensions. The focusing performances (length of focus line, diameter of the spot focus) are easily modulable by changing the pumping power. This capability can be used to find the best tradeoff between the long focusing action and the achievement of on-axis high light intensity. 

Calculations and experiments have shown that with the proper choice of the focusing lens, our device can reach performances close to those exhibited by other more refined tools (for instance, reflective axicons or metasurfaces), with the advantage of an easier and inexpensive fabrication process. Indeed, at the probe wavelength of 405 nm, we were able to produce a beam with a radial FWHM of 0.59 μm, and a FWHM depth of focus of about 90 μm. The present proof-of-concept can be designed in a more compact structure by substituting the bulky objective with a micro lens, such as a GRIN rod lens, available on the market even in sub-millimeter lengths.

## Figures and Tables

**Figure 1 micromachines-13-02083-f001:**
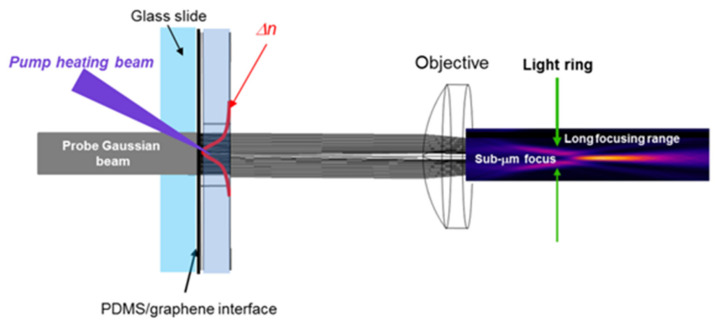
Working principle of the PDMS/SLG thermo-optical device. The absorbed pump heats up the PDMS locally, creating a refractive index profile Δn characteristic of a negative axicon. The input beam is focused into a light ring by an objective. The successive propagation produces the sub-micrometric LFRB.

**Figure 2 micromachines-13-02083-f002:**
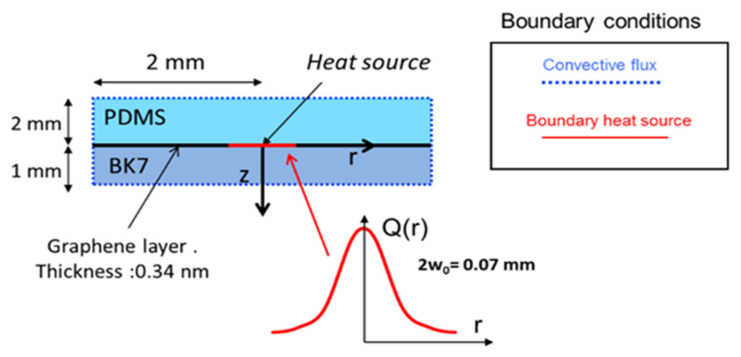
Geometry of the thermo-optical model, completed with the boundary conditions used in the COMSOL code. The convective parameter is h = 10 W/m^2^K.

**Figure 3 micromachines-13-02083-f003:**
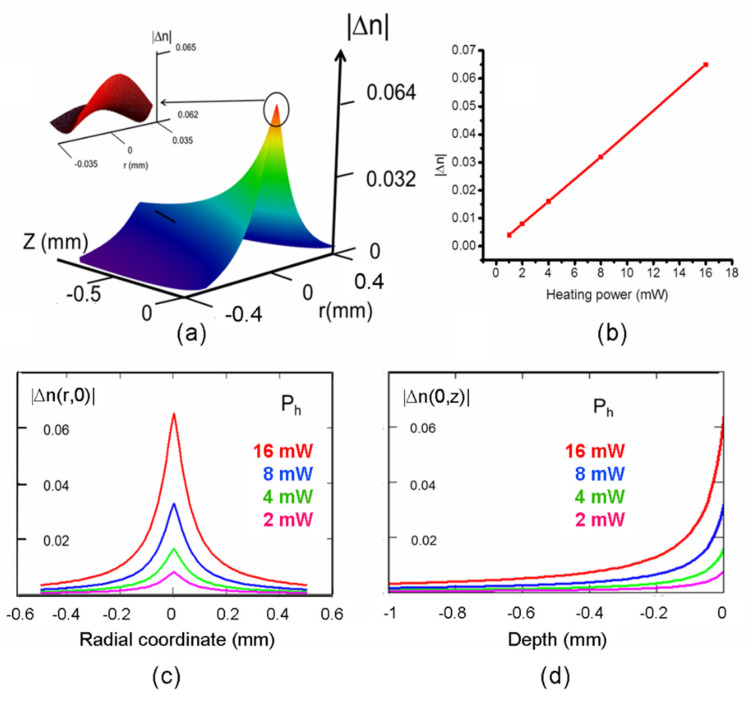
Results of the thermo−optical modeling. (**a**) Spatial variation of the refractive index due to the thermo-optical effect. In the inset, the apex distribution is shown. (**b**) Variation of |Δn(0,0)| with the heating power. (**c**) Radial distribution of the modulus on the graphene layer surface for several heating powers and (**d**) its axial distribution for the same powers.

**Figure 4 micromachines-13-02083-f004:**
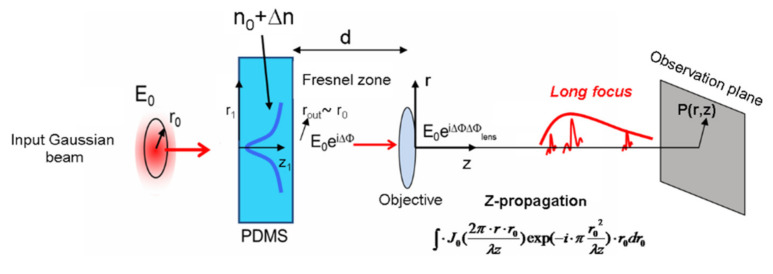
Propagation of the electromagnetic field through the optical chain. The input field E_0_ acquires two phase delays, ΔΦ and ΔΦ_lens_, due to the refractive index gradient in the PDMS and the lens, respectively. The objective is located at distance *d* from the sample of about 15 mm. The beam is finally transformed into a long focusing range beam. See text for further details.

**Figure 5 micromachines-13-02083-f005:**
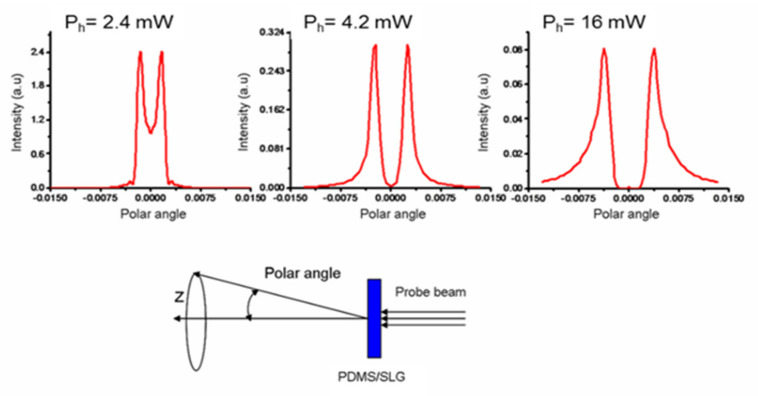
Upper part: Theoretical far field at three different heating powers P_h_ at the wavelength of 633 nm. Lower part: definition of the polar angle.

**Figure 6 micromachines-13-02083-f006:**
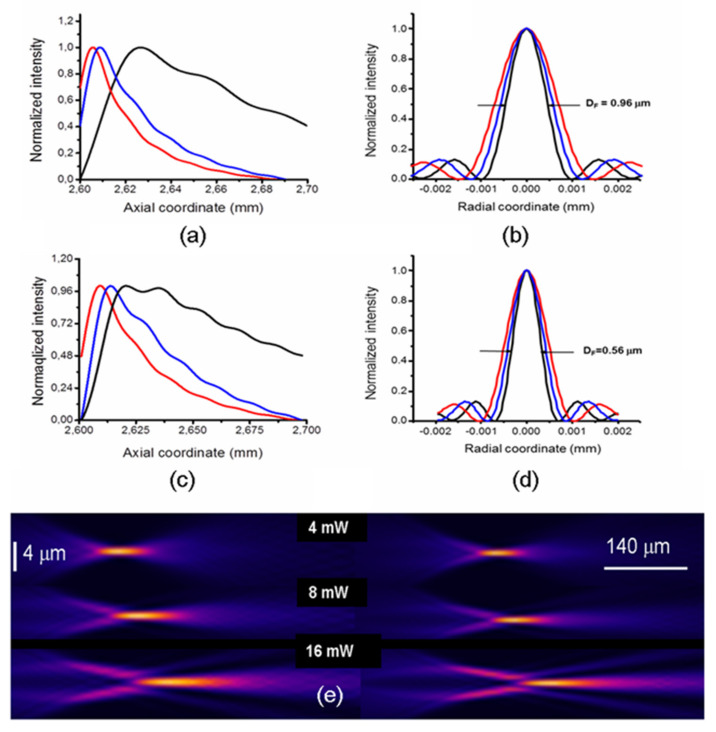
(**a**–**d**) Theoretical normalized light distributions beyond the focusing optics (focal length, origin of the x-axis = 2.6 mm) for three heating powers P_4_ = 4 mW, 8 mW, 16 mW. Upper row (**a**,**b**): *λ*_1_ = 633 nm; lower row (**c**,**d**): *λ*_2_ = 405 nm. (**a**,**c**) Axial and (**b**,**d**) radial intensities. In (**b**,**d**), only the radial FWHMs relative to P_h_ = 16 mW are reported. The other parameters are listed in Table 1 for a simpler comparison. (**e**) Computed light intensities for the same heating powers for two wavelengths *λ*_1_ (left side) and *λ*_2_ (right side). The light propagates from left to right. Notice the converging ring on the left of each focal line that evolves in the long range focus.

**Figure 7 micromachines-13-02083-f007:**
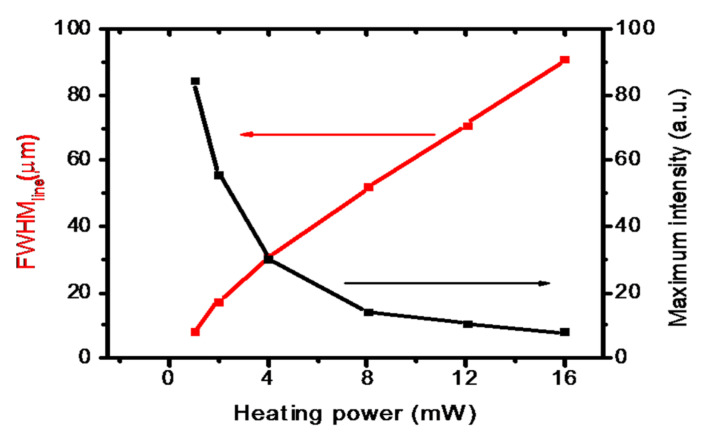
Focal line length (FWHM_line_) and maximum on-axis intensity vs. the heating power P_h_ for *λ*_2_ = 405 nm. The focal length of the focusing lens is 2.6 mm. The increase in FWHM_line_ is accompanied by a reduction in the intensity. See text for details.

**Figure 8 micromachines-13-02083-f008:**
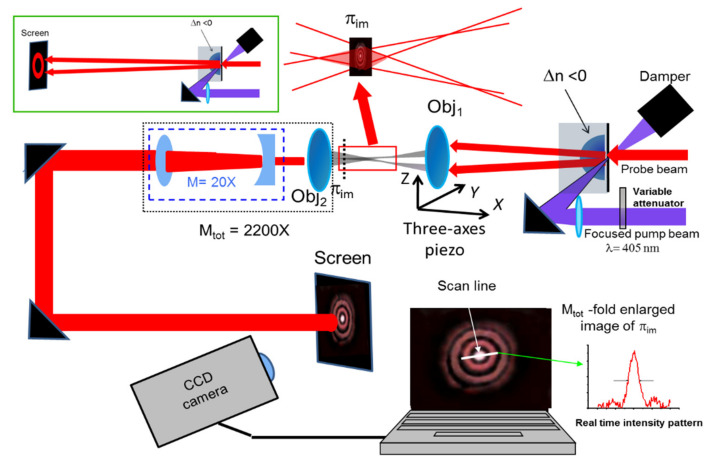
Experimental apparatus. The input probe beam travels through the heated PDMS, which modifies and transmits it to the focusing objective Obj_1_ (focal length 2.6 mm), mounted on a three-axis piezo. The focusing action determines the formation of a superposition volume (evidenced in the red box and exploded in the upper part) where the long focus develops. The section of the focus located on the plane π_im_ is conjugated onto the observation screen by a 2-stage optical magnifier, composed of an objective Obj_2_ (focal length 2.6 mm, measured magnification 110x, and a 20x Galilean telescope. The focus image, magnified 2200 times, is then projected onto the screen and imaged by a CCD camera. An image processing software permits checking the image intensity on a given line in real time, allowing the monitoring of the change in the spot dimensions in response to changes in P_h_, obtained with a variable attenuator. The measure of the far field annular distributions is performed without Obj_1_ and the 2200x magnification system (green frame in the upper left side).

**Figure 9 micromachines-13-02083-f009:**
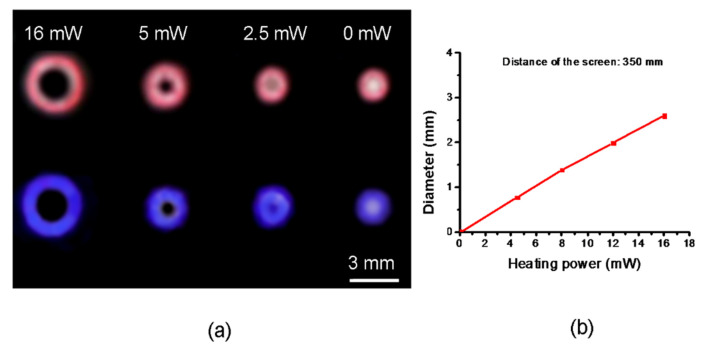
(**a**) Annular rings produced at 350 mm distance from the sample at *λ*_1_ = 633 nm (upper part of the photo) and *λ*_2_ = 405 nm (lower part) for different heating powers. (**b**) The ring diameter, intended as the distance between the maxima directly measured on a diametric scanning line (see caption of Figure 7), vs. the heating power P_h_.

**Figure 10 micromachines-13-02083-f010:**
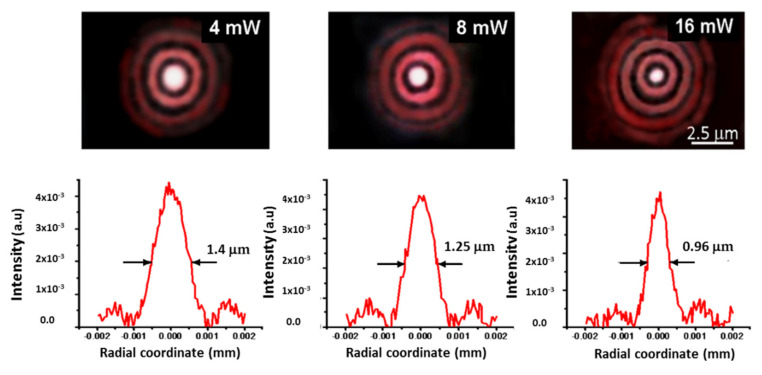
Upper part: photos of the focal LFRB spots magnified 2200 times at different heating powers at *λ*_1_ = 633 nm. The scale bar indicates the effective dimensions of the focal cross−section. Lower part: Corresponding measured diametric light distributions.

**Figure 11 micromachines-13-02083-f011:**
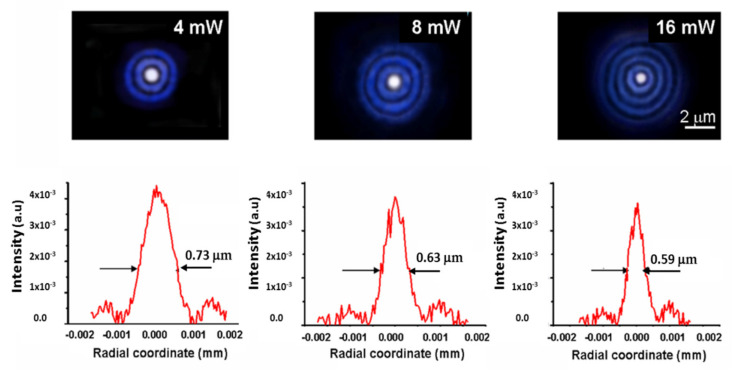
Upper part: Photos of the focal LFRB spots magnified 2200 times at different heating powers at *λ*_2_ = 405 nm. The scale bar represents the actual dimensions of the focal cross-section. Lower part: Corresponding measured diametric light distributions.

**Figure 12 micromachines-13-02083-f012:**
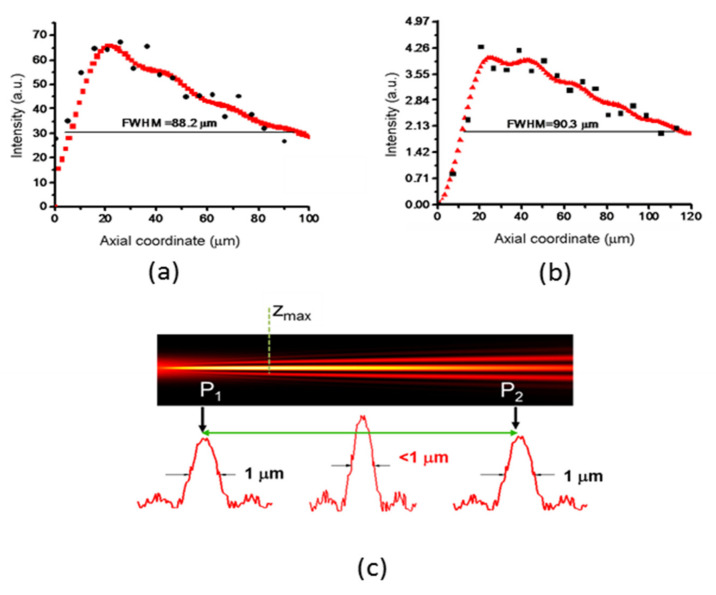
Upper part: Measured (black dots) and theoretical (red dots) light distributions in the focal lines for (**a**) *λ*_1_ = 633 nm and (**b**) *λ*_2_ = 405 nm. (**c**) Definition of the focal LFRB line P_1_P_2_, within which the FWHM spot diameter is submicrometric.

**Table 1 micromachines-13-02083-t001:** Theoretical performances of the LFRB system based on PDMS/SLG interface. Objective focal length: 2.6 mm. Probe wavelengths: 633 nm (*405* nm).

P_h_ (mW)	FWHM_line_ (μm)	D_F_ (μm)	L_μm_* (μm)	R_G_ (μm)
4	11.5 (*9.3*)	1.32 (*0.6*)	ND (*ND*)	4.4 *(1.5)*
8	26.7 (*25.5*)	1.21 (*0.6*)	ND (*21.3*)	3.6 *(1.4)*
16	88.2 (*90.3*)	0.96 (*0.5*)	8.60 *(72)*	*2.3 (1.23)*

* Axial length within which D_F_ < 1 μm.

## Data Availability

Not applicable.

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
