# Peer review of "Thermooptical PDMS-Single-Layer Graphene Axicon-like Device for Tunable Submicron Long Focus Beams"

_micromachines, 2022, doi:10.3390/mi13122083_

Round 1

Reviewer 1 Report

Comments and Suggestions for Authors

Margheri et al. reported thermo-optical PDMS-single layer graphene axicon-like device for tunable sub-micron long focus beams, which demonstrated an experimental Full Width at Half Maximum (FWHM) spot diameter of 0.59 um at the wavelength of 405 nm, with a FWHM length of the focal line higher than 90 um. In addition, the length of the focal line and the diameter of the focus can be tuned easily by the control of the pump power. The manuscript can be reconsidered after major revisions.

Some important issues listed below have to be carefully addressed:

1. The introduction part has to be reorganized and rewritten because authors have not highlighted the employed PDMS and SLG. What are the unique advantages of PDMS and SLG in this work. Why did authors choose them? Graphene is out of fashion, so authors have to give the impressive reasons why graphene can play an important role in this field.

2. Where is the structural characterization of SLG? Are authors sure that the as-fabricated graphene is single layer because graphene shows tunable properties with its layers.

3. What is the function of PDMS here? Is it possible to use other substrates (e.g., PS and PMMA) as alternatives.

4. A variety of photothermal effect-based nanostructures exist, selenium (10.1002/adfm.202003301), MXene (10.1002/adfm.202005223), bismuth nanostructures (10.1002/adfm.202007584), which can be cited and compared to further clarify the differences?

5. It would be better to give further applications such as all-optical modulation (DOI: 10.1002/adom.201700985), harmonic or Q-switched pulse generation (10.1007/s40843-020-1490-9; 10.35848/1882-0786/abf055) based on PDMS-graphene.

6. A lot of syntax errors exist in the current version, and the english presentation has to be largely polished.

Reviewer 2 Report

In this paper, the authors perform a theoretical and experimental analysis of an tunable axicon-like device based on an elastomeric polydimethylsiloxane/single layer graphene (PDMS/SLG). The device working principle is based on the phase modulation imposed to a probe beam due to the thermo-optical effect caused by the absorption of a pump beam by the SLG. By focusing the phase-modified beam with an objective lens, the authors reached a FWHM spot diameter of 0.59 um with a FWHM depth of focus of about 90 um at the wavelength of 405 nm. Their results on long focusing range beams with submicron focal spots are compatible to those achieved by devices with more sofisticated designs and phase-modulation strategies. Moreover, their device show broadband response and tunability. The authors provided a thorough characterization of the proposed device, disclosing the details of the modeling and design process, fabrication and experimental proof-of-concept. The advantages and drawbacks of their device are clearly stated, and properly compared with similar devices reported in the literature. The manuscript is well written, the methods are adequate and, overall, it introduces an advance in the field of long focus beams. Given these reasons, I recommend to accept the manuscript in the present form.

Round 2

Reviewer 1 Report

Authors have to largely enhance the figure resolution in Figure 1,2,5 and 6. 

The presentation has to be largely polished, and a lot of errors have to be carefully checked and corrected, such as space before and after "=", the uniformity of wavelength except the first appearance, etc.

Authors have to carefully address these, or I think that the manuscript can not meet the requirement for publication.
